# Cardiorespiratory Adaptation to Short-Term Exposure to Altitude vs. Normobaric Hypoxia in Patients with Pulmonary Hypertension

**DOI:** 10.3390/jcm11102769

**Published:** 2022-05-14

**Authors:** Simon R. Schneider, Mona Lichtblau, Michael Furian, Laura C. Mayer, Charlotte Berlier, Julian Müller, Stéphanie Saxer, Esther I. Schwarz, Konrad E. Bloch, Silvia Ulrich

**Affiliations:** 1Department of Pulmonology, University Hospital Zurich, Rämistrasse 100, 8091 Zurich, Switzerland; simonrafael.schneider@usz.ch (S.R.S.); mona.lichtblau@usz.ch (M.L.); michael.furian@usz.ch (M.F.); laurachmayer@gmail.com (L.C.M.); charlotte.berlier@usz.ch (C.B.); julian.mueller2@usz.ch (J.M.); stephanie.saxer@usz.ch (S.S.); estherirene.schwarz@usz.ch (E.I.S.); konrad.bloch@usz.ch (K.E.B.); 2Department of Health Sciences and Medicine, University of Lucerne, Frohburgstrasse 3, 6005 Lucerne, Switzerland

**Keywords:** normobaric hypoxia, hypobaric hypoxia, high altitude, pulmonary hypertension, chronic thromboembolic pulmonary hypertension

## Abstract

Prediction of adverse health effects at altitude or during air travel is relevant, particularly in pre-existing cardiopulmonary disease such as pulmonary arterial or chronic thromboembolic pulmonary hypertension (PAH/CTEPH, PH). A total of 21 stable PH-patients (64 ± 15 y, 10 female, 12/9 PAH/CTEPH) were examined by pulse oximetry, arterial blood gas analysis and echocardiography during exposure to normobaric hypoxia (NH) (F_i_O_2_ 15% ≈ 2500 m simulated altitude, data partly published) at low altitude and, on a separate day, at hypobaric hypoxia (HH, 2500 m) within 20–30 min after arrival. We compared changes in blood oxygenation and estimated pulmonary artery pressure in lowlanders with PH during high altitude simulation testing (HAST, NH) with changes in response to HH. During NH, 4/21 desaturated to SpO_2_ < 85% corresponding to a positive HAST according to BTS-recommendations and 12 qualified for oxygen at altitude according to low SpO_2_ < 92% at baseline. At HH, 3/21 received oxygen due to safety criteria (SpO_2_ < 80% for >30 min), of which two were HAST-negative. During HH vs. NH, patients had a (mean ± SE) significantly lower PaCO_2_ 4.4 ± 0.1 vs. 4.9 ± 0.1 kPa, mean difference (95% CI) −0.5 kPa (−0.7 to −0.3), PaO_2_ 6.7 ± 0.2 vs. 8.1 ± 0.2 kPa, −1.3 kPa (−1.9 to −0.8) and higher tricuspid regurgitation pressure gradient 55 ± 4 vs. 45 ± 4 mmHg, 10 mmHg (3 to 17), all *p* < 0.05. No serious adverse events occurred. In patients with PH, short-term exposure to altitude of 2500 m induced more pronounced hypoxemia, hypocapnia and pulmonary hemodynamic changes compared to NH during HAST despite similar exposure times and P_i_O_2_. Therefore, the use of HAST to predict physiological changes at altitude remains questionable. (ClinicalTrials.gov: NCT03592927 and NCT03637153).

## 1. Introduction

Because of the easy access to mountain areas, mountain tourism, the many populated high altitude areas, and the affordable costs of long-distance flights, the number of people exposing themselves to hypoxia is increasing [1]. Hence, patients with cardio-vascular or respiratory disorders such as pulmonary arterial and chronic thromboembolic pulmonary hypertension (PAH/CTEPH, summarized as PH) also wish to undergo high altitude journeys and air travel for various purposes. Hypobaric hypoxia (HH) increases with increasing altitude and decreasing barometric pressure (Pbar) resulting in an increasing prevalence of altitude-related adverse health effects (ARAHE) in healthy but particularly in patients with cardiorespiratory diseases [2,3,4,5,6]. The lower Pbar resulting in lower alveolar, blood- and tissue oxygenation leads to immediate hypoxic pulmonary vasoconstriction (HPV) that may put PH-patients at a particularly high risk for a further increase in pulmonary artery pressure (PAP) and vascular resistance (PVR) in response to hypoxia [7]. Comprehensive data on the acute effects of exposure to normobaric hypoxia (NH) in PH have recently been published [8,9] and effects of an exposure to HH of PH patients during a daytrip to altitude (2500 m, Mount Saentis) have been described [10]. However, adequate counselling of PH-patients with regard to altitude or air travel remains challenging, as risk factors and predictors for ARAHE and potential need of supplemental oxygen are largely unknown [8,9,10,11]. In patients with chronic obstructive pulmonary disease (COPD), it is recommended to use supplemental oxygen during air travel when in-flight partial pressure of O_2_ (PaO_2_)-levels drop < 6.6 kPa, albeit this recommendation is not based on robust evidence and the critical blood oxygen level inducing ARAHE is unknown [12,13]. To evaluate the patient’s risk for ARAHE during air travel, some experts recommend performing the high altitude simulation test (HAST) [11,12,14]. During the HAST, patients breathe a nitrogen-enriched gas mixture with an inspired O_2_ fraction (F_i_O_2_) of 15% to simulate inflight conditions. The resulting inspired partial pressure of O_2_ (P_i_O_2_) corresponds approximately to the minimally allowed pressure in commercial aircraft cabins (pressurized to 752 hPa, equivalent to 2438 m, 8000 ft) [11]. The HAST is used for COPD-patients with borderline SpO_2_ levels < 95% near sea level. It has been shown to predict inflight oxygen saturation with varying accuracy [11,15]. However, evidence-based counseling based on HAST for upcoming air-travel is limited as its ability to predict ARAHE has not been evaluated [11,16,17]. Nevertheless, the British Thoracic Society expert group recommends supplemental inflight oxygen if SpO_2_ at the end of HAST is < 85% or PaO_2_ < 6.6 kPa) [12,13].

The aim of the present analysis was to compare changes in blood oxygenation and echocardiographically assessed PAP in response to exposure to NH during HAST to their respective changes induced by a similar exposure time to HH at the equivalent real altitude (2500 m). Furthermore, we evaluated whether outcomes from HAST predict the occurrence of ARAHE within the first hours at 2500 m altitude.

## 2. Materials and Methods

### 2.1. Subjects

Adult outpatients with precapillary PH due to PAH or distal CTEPH diagnosed in accordance with current guidelines [18] were recruited in the reference centre for PH of the University Hospital Zurich and included upon written informed consent. Patients were clinically stable, living at low altitude < 1000 m, had a resting PaO_2_ > 7.3 kPa and a PaCO_2_ < 6.5 kPa. Patients exposed to >1500 m for >3 nights during the previous 4 weeks, pregnant- or breastfeeding women or patients with relevant comorbidities (PH of other groups such as from heart- or lung disease or miscellaneous reasons, severe concomitant diseases such as cancer, psychological disorders and illicit drugs or alcohol abuse) were excluded. This subgroup analysis where patients participated in two already published randomized cross-over trials investigating the effect of NH vs. placebo (air at 470 m) and the effect of a daytrip to HH at 2500 m vs. 470 m investigates the predictive value from NH to altitude exposure and physiological differences between NH and HH. Part of these data, mainly from the NH group, have been previously published [10]. The studies followed the principles of the declaration of Helsinki, was approved by Cantonal Ethics Committee Zurich (KEK 2018-00455), and registered at ClinicalTrials.gov (NCT03592927 and NCT03637153).

### 2.2. Design and Interventions

The methods of the trials evaluating arterial blood gas and hemodynamic changes induced by NH and a daytrip to HH, respectively, vs. room air at 470 m at rest and during exercise have been previously reported [9,10]. The present analysis focuses on the difference induced by NH at low altitude (simulated altitude) vs. HH at high altitude (2500 m) in the same patients during a comparable exposure time and inspired partial pressure of oxygen (P_i_O_2_). NH with an F_i_O_2_ of 15% was applied via a tightly-fitted facemask with a non-rebreathing two-way valve (Hans-Rudolph, Shawnee, KS, USA) connected to an ambient air-nitrogen mixing device (AltiTrainer, Lausanne, Switzerland) during 30–45′ as described [9]. Measurements during HH were performed 20–30′ after patients arrived at 2500 m at Mount Saentis by cable car (10 min ascent) [10].

### 2.3. Assessments

Baseline measurements included demographics, PH-classification, current medication, and New York Heart Association (NYHA) functional class.

The following assessments were performed 20–30′ after exposure to NH during a HAST and 20–30′ after arrival on Mount Säntis at 2500 m by cable car:

Transthoracic echocardiography was performed in supine position at rest (CX50, Philips Respironics, Horgen, Switzerland) according to current guidelines [19]. The maximal tricuspid regurgitation velocity was assessed by Doppler to derive the tricuspid regurgitation pressure gradient (TRPG) with the simplified Bernoulli equation ΔP = 4 × Vmax2, which served as surrogate for the PAP. Fractional area change (FAC), tricuspid annular plane systolic excursion (TAPSE), stroke volume (SV) (=the left ventricular outflow tract velocity time integral (LVOT VTI) × π × (LVOT diameter/2)2), and cardiac output (CO) (=SV × heart rate (HR)) were measured. We estimated the resting right atrial pressure (RAP) from the respiratory variability of the inferior vena cava. The systolic PAP (SPAP) was calculated as TRPG + RAP and mean PAP as 0.61 × systolic PAP + 2 [20]. Pulmonary artery wedge pressure (PAWP) was computed by 1.24 × (E/E′) +1.9 [21]. PVR was calculated by (mean PAP–PAWP)/CO [19,22,23].

Arterial blood was sampled from the radial artery at rest and immediately analyzed (ABL90 Flex, Radiometer GmbH, Switzerland). The oxygen content (CaO_2_) was calculated by the following formula: (Hb × 1.36 × (SaO_2_/100)) + ((7.5 × PaO_2_) × 0.0031) and was multiplied by CO for oxygen delivery (DaO_2_) [24].

HR, breathing rate (BR) and fingertip SpO_2_ were continuously recorded by Alice-PDX^®^ (Philips Respironics, Horgen, Switzerland).

The 10-cm visual analogue scale was used for assessing general wellbeing and dyspnea at the end of the echocardiography in upright position [25].

### 2.4. Index and Reference Test

HAST, index test: We applied the same HAST criteria for in-flight supplemental oxygen recommendations as proposed by the British Thoracic Society [26,27,28]. Therefore, a positive HAST was defined as a SpO_2_ < 85% or a PaO_2_ < 6.6 kPa after 20 min exposure to NH. 

ARAHE, reference test: Patients with severe dyspnea or general discomfort, new onset angina, cardiac deterioration (arrhythmia, dizziness), symptoms of acute mountain sickness or an SpO_2_ < 80% for >30 min or <75% for >15 min were defined as experiencing an ARAHE, and resulted in the indication for oxygen therapy with the aim of achieving an SpO_2_ > 90% and accompanied return to low altitude [6,29].

### 2.5. Data Analysis and Statistics

Baseline characteristics are summarized as means ± standard deviation (SD) or number (proportions). Continuous physiological data from Alice-PDX^®^ (following 20–30 min of exposure to NH or HH) were imported to LabChart Data analysis software, version 8 and averaged over 30 s. Data are summarized as means ± standard errors and the difference between NH and HH shown as mean-difference with 95% confidence intervals. Mixed-effects linear regression models with fixed effects (Time, Age, Gender and NYHA-class) and calculated average marginal effects generated by Stata^®^’s linear combinations of parameters (lincom) were used. The sensitivity and specificity to predict ARAHE by BTS-criteria for HAST at low altitude were calculated. A receiver operator-characteristic curve (ROC) including area under the curve (AUC) for the best predicting cut-off for ARAHE of SpO_2_ and PaO_2_ during HAST were calculated. A two-sided *p*-value <0.05 was considered as statistically significant. Statistical analysis was performed with Stata statistical software, version 16.

## 3. Results

### 3.1. Study Population

Based on the recruitment structure from the initial trial examining HA-exposure within a daytrip, 124 patients with PH were assessed for eligibility. A total of 21 patients (12 PAH, 9 distal CTEPH, 10 female, age 64 ± 15 years) participated in both trials. Patients performed a HAST within 42 ± 25 days before or after the daytrip to 2500 m (HH) (Figure 1). Demographic characteristics are summarized in Table 1 [9,10].

#### 3.1.1. Differences between Exposure to Normobaric vs. Hypobaric Hypoxia

Differences between NH and HH are shown in Table 2 and Figure 2. Linear mixed-effect regression models revealed that HR and BR did not significantly differ between NH and HH. Pulse oximetry and arterial blood gas analysis revealed a significantly lower blood oxygenation including PaO_2_, SaO_2_ and oxygen content (CaO_2_). PaCO_2_ and bicarbonate (HCO3-) were also significantly lower in HH compared to NH, whereas pH was significantly higher (Figure 2). TRPG was significantly higher during HH compared to NH, which also led to a significant difference in SPAP and the TAPSE/SPAP ratio. Other echocardiographic measurements did not differ significantly between NH and HH. Patient-reported visual analogue scales, dyspnea-related symptoms and general well-being, did not differ among NH compared to HH (Table 2, Figure 3).

#### 3.1.2. HAST and Baseline Measures to Predict Supplemental Oxygen at 2500 m

According to the BTS-recommendations for air travel in respiratory diseases, 12 out of 21 (57%) participants would have qualified for supplemental oxygen due to low SpO_2_ (<92%) on room air at 470 m and 8 had an indication for further testing by the HAST due to resting SpO_2_ values between 92–95% or PaO_2_ < 9.3 kPa [26,27]. Based on findings from the HAST, 4 (19%) would have qualified for supplemental oxygen at altitude. At 2500 m, 3 (14%) patients (1 HAST-positive and 2 HAST-negative) actually experienced an ARAHE and thus received supplemental oxygen (Figure 4). Thus, the sensitivity of the HAST for experiencing an ARAHE at altitude was 33.3% in terms of SpO_2_- and PaO_2_-levels. The specificity of the SpO_2_- and PaO_2_-criteria of the HAST for ARAHE was 83.3% and 82.4%, respectively. The best predictive parameter assessed within these studies for an ARAHE at altitude was the baseline NYHA class with an AUC of 0.85 (Figure 5). Being in NYHA class 3 revealed a sensitivity for ARAHEs of 66.6% and a specificity of 88.8%.

## 4. Discussion

In this unique study, we compared hemodynamic and arterial blood gas changes in patients with pulmonary vascular disease during a comparable exposure time to NH at lowland (simulated altitude) and HH at altitude (2500 m). We found that HH decreased blood oxygenation to a significantly higher extent and resulted in a higher increase in PAP. The predictive value of low altitude baseline SpO_2_ or the SpO_2_ and PaO_2_ at the end of a HAST to predict ARAHE was very low. Our results strongly indicate that physiological adaption during NH vs. HH differ and the value of the HAST to predict ARAHE is limited in patients with PH.

Several researchers made efforts to find predictors to assess fitness-to-fly for patients with respiratory diseases [11,14,15,30]. During commercial long-distance aircraft travelling, the minimal allowed Pbar is 752 hPa, equivalent to an altitude of around 2500 m. Several studies focused on fitness-to-fly but hardly any on fitness-for-altitude travel for patients with cardiorespiratory diseases. [31,32]

Similarities between NH and HH were first described in the Equivalent Air Altitude Model by Paul Bert (1878). This model explains identical physiological responses in both conditions on the human body provided that the P_i_O_2_ remains unaltered. In other words, a reduced F_i_O_2_ with low-altitude ambient Pbar was suggested to be comparable to a normal F_i_O_2_ (20.9%) and a reduced Pbar as at altitude. However, this proposal resulted in numerous criticisms since the observed responses to NH often differed from HH [32,33,34].

In the current study, 3 out of 21 participants revealed severe hypoxemia during HH and had to receive oxygen therapy according to predefined safety criteria. The predictive value of the HAST was hereby very low, making this measure useless for predicting ARAHE [35].

Several previous studies, most of which had small sample sizes, focused on identifying patients with respiratory disease at risk for air travel or on predicting a low PaO_2_ at altitude. Most of them assessed fitness-to-fly in patients with COPD, who later underwent air-travel with or without inflight oxygen. In line with our results, the majority of studies resulted in the conclusion that the prediction of HAST for ARAHE is limited [28,36,37]. In the absence of robust evidence, HAST is still commonly used to assess fitness-to-fly.

An explanation for the inability of the HAST to predict changes during HH might be that NH is not an experimental set-up directly comparable to HH and that the setting of breathing hypoxic gas mixtures through tubes results in different changes compared to spontaneous breathing at HH. In our study, we exposed patients with pulmonary vascular disease to NH (F_i_O_2_: 15%) and HH (2500 m) during a similar exposure time in a randomized-cross-over design and indeed found several significant differences between the associated changes in estimated PAP and oxygenation. We observed a significant lower SpO_2_, PaO_2_, and PaCO_2_ as well as more pronounced respiratory alkalosis at altitude compared to NH (Table 2, Figure 2). This lower blood oxygenation despite more pronounced hyperventilation and thus higher respiratory drive as indicated by the lower PaCO_2_ at altitude, was associated with a significantly higher PAP under HH at altitude compared to equivalent NH. In addition, we found hints for reduced ventriculo-arterial coupling as indicated by the lower TAPSE/SPAP ratio.

Current literature contains plenty of criticism of the Equivalent Air Altitude model, since acute mountain sickness (AMS) and other physiological parameters differed between the same hypoxic inspired oxygen partial pressure (P_i_O_2_) at different barometric pressures [32].

In 1982, Grover et al. reported that a reduction in Pbar alone, without a change in P_i_O_2_, may reduce peripheral chemoreceptor sensitivity, but a plausible mechanism was not described [38]. Savourey, Tucker and Loeppky observed a decrease in ventilation at HH versus NH with a lower tidal volume and higher respiration rate (fast but shallow). Loeppky raised the notion that the lower gas density at altitude is responsible for a smaller CO_2_ production, which could reduce the ventilation and would also reduce the work of breathing and therefore the alveolar dead space [32]. However, in our study population, PaCO_2_ was lower at altitude, indicating an increased ventilation mainly due to a higher respiratory rate (tendency towards a higher breathing rate compared to NH with a borderline significance-level (*p* = 0.056)). However, minute ventilation was not measured in our study.

Data from 9 trained healthy men reported by Loeppky et al. revealed no difference in SpO_2_ between NH and HH up to 10 h, whereas the study by Savourey et al. including 18 healthy men revealed a lower blood oxygenation during 40 min at HH vs. NH in line with our results in PH-patients [32,39]. Part of the difference between these studies and our results may be explained due to the fact that we exposed PH patients to HH at real altitude after ascent by cable car, a setting different to experimental hypobaric chamber conditions and that our subjects were significantly older, half were females and all suffered from chronic pulmonary vascular disease.

Another potential mechanism potentially explaining a difference between NH and HH as found in our PH-patients may be the difference in N2-movements resulting in lower PaO_2_ and PaCO_2_ in HH as previously postulated [32].

The lower blood oxygenation during HH vs. NH in our study was associated with an elevated PAP, presumably due to enhanced hypoxic pulmonary vasoconstriction [8,40]. Driven by the increased PAP at altitude, the TAPSE/SPAP was reduced, which may indicate a slightly more impaired ventricular–arterial coupling at 2500 m vs. NH. However, we believe that the exposure time was too short to influence right-heart stiffness [23].

### Limitations

The sample size was rather small. However, the cross-over design allows a high statistical power with a low sample size, and this study demonstrated a significant difference in the main outcomes of interest, PaO_2_ and TRPG, with high statistical power (99%). As we investigated HH at real altitude, the setting at altitude was different and participants had to travel by cable car with a short break during ascent on 1300 m while waiting for the gondola, thus the exposure time to HH might have been slightly less acute/slower. However, this reflects reality for any altitude stay or air-travel and this real-life setting strengthens the conclusions from our findings at HH.

## 5. Conclusions

Hypobaric hypoxia exposure at real altitude results in a significantly lower blood oxygenation and higher PAP compared to simulated altitude using HAST during comparable exposure time and at comparable P_i_O_2_ at real and simulated altitudes.

The value of the HAST protocol to predict ARAHE was very weak, and thus its use to counsel patients in regard to risk of ARAHE during air travel or altitude stays is questionable.

## Figures and Tables

**Figure 1 jcm-11-02769-f001:**
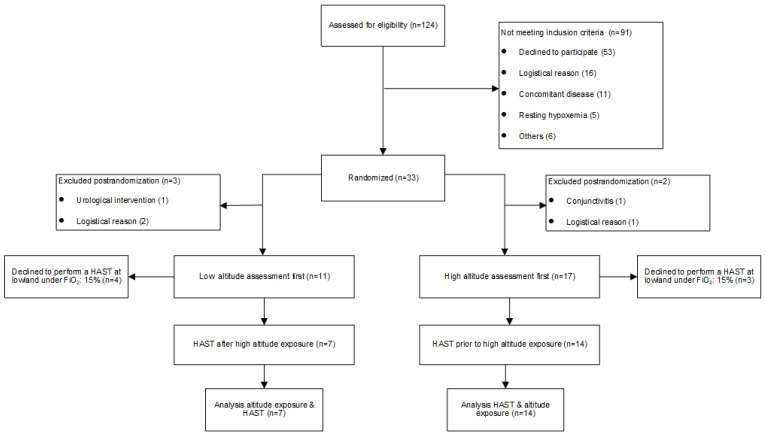
Flow chart: HAST = High altitude simulation test.

**Figure 2 jcm-11-02769-f002:**
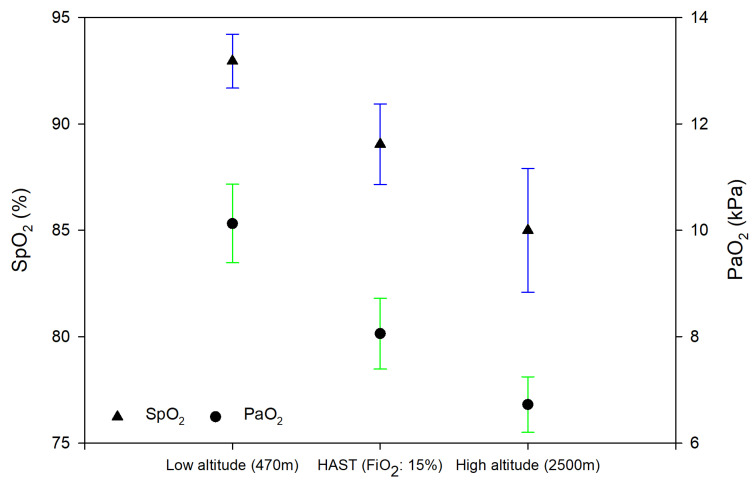
SpO_2_ (triangles, blue) and PaO_2_ (circles, green) in normobaric normoxia (470 m), normobaric hypoxia (470 m) and at high altitude (2500 m). HAST = High altitude simulation test. Measurements are summarized as means and 95% Confidence Intervals. Data are computed from a mixed-effect regression model with fixed effects variables as time, age, gender and New York Heart Association class.

**Figure 3 jcm-11-02769-f003:**
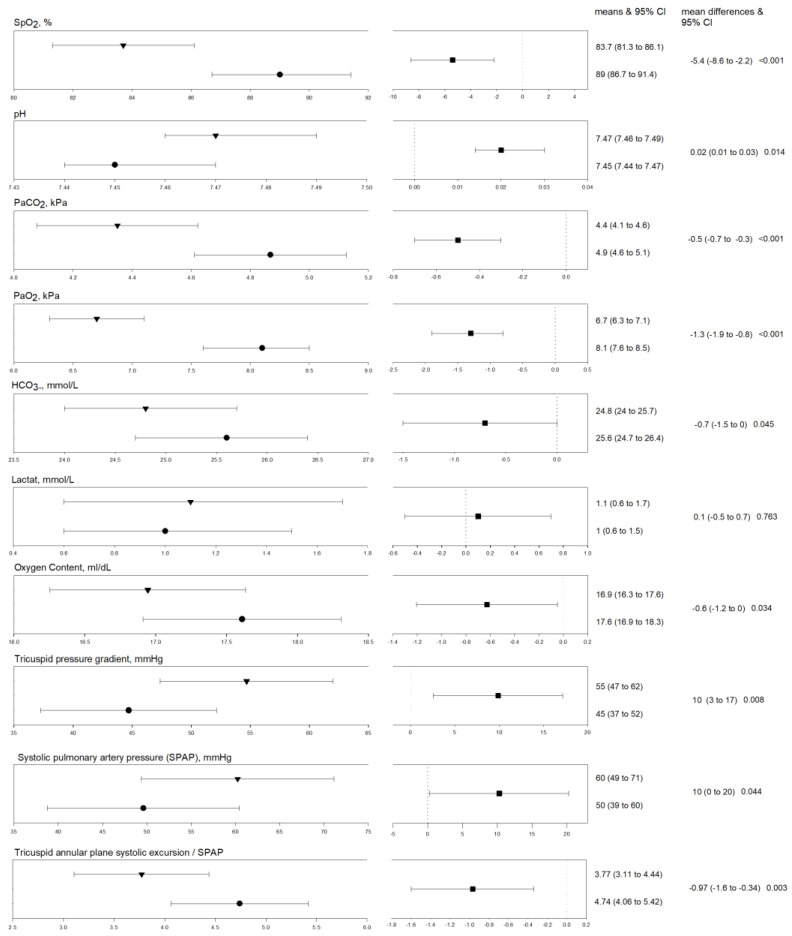
Physiological outcomes after 20–30 min under normobaric (F_i_O_2_: 15%) vs. hypobaric hypoxia (2500 m). Triangles and circles indicate mixed-effect model derived means and 95% confidence intervals (CI) of outcomes during normobaric hypoxia (F_i_O_2_: 15%) and hypobaric hypoxia (2500 m) after 20–30 min of exposure. Fixed effects variables are Time, Age, Gender and New York Heart Association class and random effect is the subject ID. Cubes indicate the mixed-effect model derived mean-differences with associated 95% CI.

**Figure 4 jcm-11-02769-f004:**
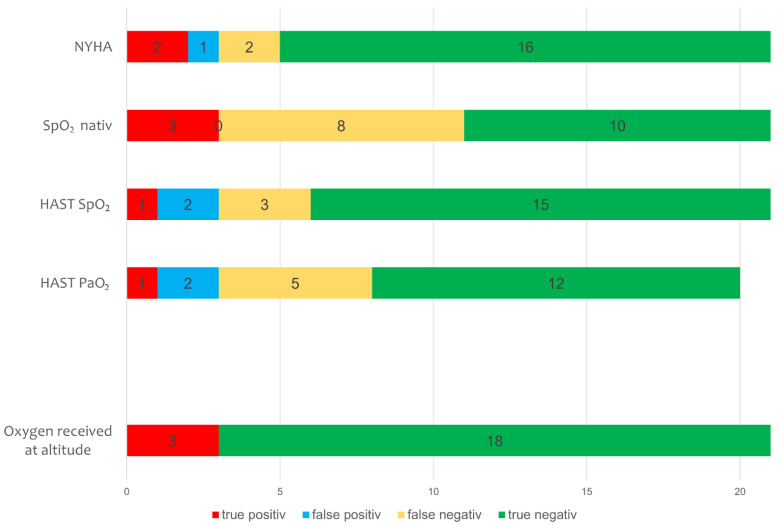
Predictions for oxygen at altitude. NYHA = New York Heart Association class, HAST = High altitude simulation test, numbers in bar represent absolute numbers. PaO_2_ (n = 20).

**Figure 5 jcm-11-02769-f005:**
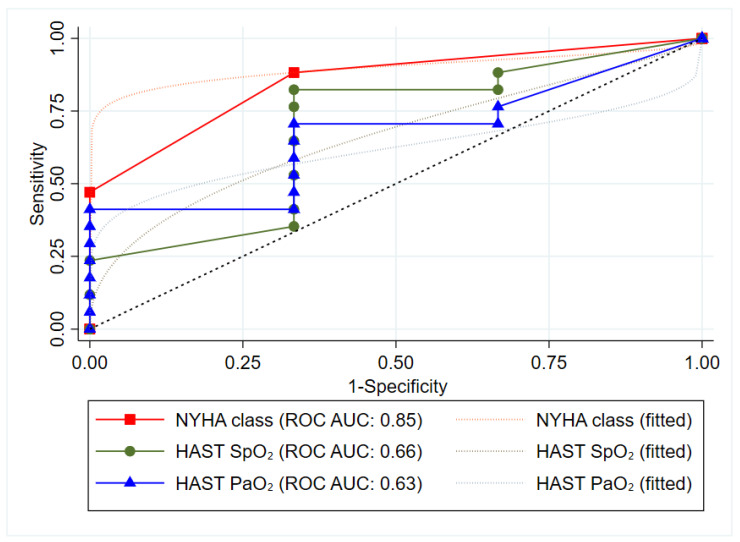
Receiver Operator Characteristic Curve for altitude related adverse health effects at 2500 m. HAST = High altitude simulation test including an exposure to normobaric hypoxia F_i_O_2_: 15%, NYHA = New York Heart Association class, ROC = Receiver Characteristic Operator Curve, AUC = Area under Curve.

**Table 1 jcm-11-02769-t001:** Baseline characteristics.

Participants/women (%)	21/10 (48)
Age, years	64 ± 15
Body mass index, kg/m^2^	25.6 ± 3.8
Pulmonary hypertension classification	
1. Pulmonary arterial hypertension	12 (58)
1.1. idiopathic	10 (48)
1.4.1. connective tissue disease	1 (5)
1.4.3. portopulmonary hypertension	1 (5)
4. Chronic thromboembolic pulmonary hypertension	9 (43)
6-min walk distance, m	538 ± 94
New York Heart Association functional class I, II, III	8 (38), 9 (43), 4 (19)
N-terminal pro brain natriuretic peptide, ng/l	427 ± 620
Incremental ramp cycle exercise, Watt	114 ± 36
Maximal oxygen uptake, ml/min/kg	18.2 ± 3.9
Resting arterial partial pressure of oxygen, kPa	10.1 ± 1.6
Mean pulmonary arterial pressure, mmHg *	42 ± 11
Pulmonary vascular resistance, WU *	6 ± 3
PH targeted therapy	
Endothelin receptor antagonist	14 (67)
Phosphodiesterase-5 inhibitor including Soluble guanylate cyclase stimulators	9 (43)
Soluble guanylate cyclase stimulators	2 (10)
Prostacyclin-receptor agonist or prostacyclin	2 (10)
Combination therapy	8 (38)

Data shown as number (%) or mean ± SD, * = assessed during last right heart catheter., WU = Wood unit.

**Table 2 jcm-11-02769-t002:** 20–30 min normobaric hypoxia (F_i_O_2_ 15%) vs. hypobaric hypoxia (2500 m).

Parameter	F_i_O_2_: 15% (Mean ± SE)	Altitude (2500 m) (Mean ± SE)	Mean Difference (95% CI)	*p*-Value
Peripheral oxygen saturation, %	89 ± 1.2	83.7 ± 1.2	−5.4 (−8.6 to −2.2)	<0.001
pH	7.45 ± 0.01	7.47 ± 0.01	0.02 (0.00 to 0.03)	0.014
Partial pressure of carbon dioxide, kPa	4.9 ± 0.1	4.4 ± 0.1	−0.5 (−0.7 to −0.3)	<0.001
Partial pressure of oxygen, kPa	8.1 ± 0.2	6.7 ± 0.2	−1.3 (−1.9 to −0.8)	<0.001
Hydrogen carbonate, mmol/L	25.6 ± 0.4	24.8 ± 0.4	−0.7 (−1.5 to 0.0)	0.045
Lactate, mmol/L	1 ± 0.2	1.1 ± 0.3	0.1 (−0.5 to 0.7)	0.763
Arterial oxygen saturation, %	90.5 ± 1.2	83.7 ± 1.2	−6.8 (−9.6 to −4.0)	<0.001
Arterial oxygen content, mL/dL	17.6 ± 0.4	16.9 ± 0.4	−0.6 (−1.2 to 0.0)	0.034
Heart rate, min^−1^	71 ± 3	69 ± 3	−1.2 (−9 to 6.6)	0.762
Breathing rate, min^−1^	16 ± 1	19 ± 1	3 (0 to 7)	0.056
Right atrial pressure, mmHg	4 ± 1	5 ± 1	1 (0 to 1)	0.160
Tricuspid regurgitation pressure gradient (TRPG), mmHg	45 ± 4	55 ± 4	10 (3 to 17)	0.008
Systolic pulmonary arterial pressure (SPAP), mmHg	50 ± 6	60 ± 6	10 (0 to 20)	0.044
Stroke volume, mL	73 ± 4.4	74.3 ± 4.3	0.8 (−9.4 to 11)	0.876
Cardiac output (CO), L/min	5.1 ± 0.4	5.1 ± 0.4	0 (−0.9 to 0.8)	0.920
Oxygen delivery, mL/min	906.6 ± 66.9	854.6 ± 65.6	−44.2 (−192.9 to 104.6)	0.560
Pulmonary vascular resistance, WU	4.3 ± 2.9	5.4 ± 2.6	1.4 (−6.3 to 9.1)	0.718
TRPG/CO, WU	8.4 ± 2	11.3 ± 2	2.8 (−2.5 to 8.2)	0.302
Tricuspid annular plane systolic excursion (TAPSE), cm	2 ± 0.1	2 ± 0.1	0 (−0.2 to 0.1)	0.760
TAPSE/SPAP ratio	4.74 ± 0.35	3.77 ± 0.34	−0.97 (−1.6 to −0.34)	0.003
Fractional area change, %	32 ± 2	30 ± 2	−2 (−6 to 2)	0.422
Visual analog scale (general wellbeing), cm	8.5 ± 0.4	8.7 ± 0.4	0.17 (−0.78 to 1.12)	0.730
Visual analog scale (dyspnea), cm	8.2 ± 0.5	8.4 ± 0.5	0.21 (−0.83 to 1.24)	0.694

Data are computed from a mixed-effect regression model with fixed effects variables as Time, Age, Gender and New York Heart Association class; F_i_O_2_: fraction of inspired oxygen, CI: Confidence Interval.

## Data Availability

Data available on request due to restrictions e.g., privacy or ethical. The data presented in this study are available on reasonable request from the corresponding author. The data are not publicly available due to reasons of sensitivity of human data.

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
