# Peer review of "Cardiorespiratory Adaptation to Short-Term Exposure to Altitude vs. Normobaric Hypoxia in Patients with Pulmonary Hypertension"

_jcm, 2022, doi:10.3390/jcm11102769_

Round 1

Reviewer 1 Report

This manuscript, by Schneider and colleagues, aims to compare hemodynamic and arterial blood gas changes in patients with pulmonary vascular disease during a comparable exposure time to Normobaric hypoxia (NH) at lowland (simulated altitude) and Hypobaric hypoxia (HH) at altitude (2500 m). Overall, this is an interesting and well-written study on the effects of NH and HH on blood oxygenation and hemodynamic changes in a population diagnosed with pulmonary hypertension. Nonetheless, I have major concerns regarding the originality of the data and potential duplication of already published data.

Major critics:

  1. Two articles previously published by the same group utilized the same data regarding arterial blood gas analysis and echocardiography data [Refer to Reference A, below] for the normobaric hypoxia and [B] for the hypobaric hypoxia, except for 7 patients that were excluded from the aforementioned original research articles. The duplication of information can be clearly attested when comparing the flowchart of article B (see fig. 1 from Schneider et al. 2021) and the flowchart present in the manuscript undergoing peer-review (refer to Fig. 1 from current manuscript). In order to consider this article suitable for publication, the authors must establish if the participants and data used in this study is the same as the data published in previous articles, except for those 7 participants. Until this major concern is addressed, this manuscript is not suitable for publication. 

  1. Schneider, S.R.; Mayer, L.C.; Lichtblau, M.; Berlier, C.; Schwarz, E.I.; Saxer, S.; Furian, M.; Bloch, K.E.; Ulrich, S. Effect of Normobaric Hypoxia on Exercise Performance in Pulmonary Hypertension: Randomized Trial. Chest 2021, 159, 757-771, doi:10.1016/j.chest.2020.09.004 
  2. Schneider, S.R.; Mayer, L.C.; Lichtblau, M.; Berlier, C.; Schwarz, E.I.; Saxer, S.; Tan, L.; Furian, M.; Bloch, K.E.; Ulrich, S. Effect of a daytrip to altitude (2500 m) on exercise performance in pulmonary hypertension – randomized cross-over trial. 2021, 00314-02021, doi:10.1183/23120541.00314-2021 %J ERJ Open Research.

Minor critics:

  1. In the exclusion criteria, the authors mentioned that patients with relevant comorbidities were excluded from this study. The authors should mention exactly which comorbidities do the authors consider relevant.

  1. All figures need several adjustments:
    • The resolution of all figures needs to be improved.
    • Please add a legend for figure 2 and 3 providing detailed explanation of symbols used, statistical test used and other relevant information.

Reviewer 2 Report

The authors compared the results of the High Altitude Simulation Testing (HAST) and true high altitude testing in patients with pulmonary hypertension. The authors also examined whether HAST is truly predictive of ARAHE and showed that HAST is incomplete in its evaluation. I think the authors' point of view is very unique. There are some minor limitations in the research methodology that the authors point out, but I can' t find any major problems. I agree with the authors' main arguments. However, I would like to suggest some improvements.

  1. The results presented in Table 2 are based on univariate analysis only. Have you performed these multivariate analyses? If the results show significant differences in SpO2 and PaO2 between NH and HH, the results may strengthen the authors' argument. However, the number of participants in this study may be underpowered for detection.

  1. In Figure 4 and Figure 5, the authors test whether HAST is predictive of ARAHE. Are there better predictors of ARAHE for items in Table 2 for which there is no significant difference between NH and HH? If such is available, the authors may be able to propose a better predictor of ARAHE to replace HAST.

  1. Some of the Figures have small text, which may be a distress to the reader. These could be improved. Also, the description of Table is out of the journal's description rules and should be corrected.

Round 2

Reviewer 1 Report

This manuscript, by Schneider and colleagues, aims to compare hemodynamic and arterial blood gas changes in patients with pulmonary vascular disease during a comparable exposure time to Normobaric hypoxia (NH) at lowland (simulated altitude) and Hypobaric hypoxia (HH) at altitude (2500 m). Overall, this is an interesting and well-written study on the effects of NH and HH on blood oxygenation and hemodynamic changes in a population diagnosed with pulmonary hypertension. Nonetheless, I still have major concerns regarding the originality of the data and potential duplication of already published data. In their cover letter authors have confirmed the partial use of already published data. Already published data should not be published as an original paper. I would recommend sending this manuscript as a letter to the editor to the same journal were the authors published the duplicated data.